# Steroid-Refractory Cholestatic Immune-Mediated Hepatitis Following Nivolumab Therapy in an Elderly Patient with Metastatic Melanoma: A Rare and Challenging Presentation

**DOI:** 10.3390/curroncol32120663

**Published:** 2025-11-27

**Authors:** Luis Posado-Dominguez, Jorge Feito-Perez, María Escribano-Iglesias, Miriam Bragado Pascual, Emilio Fonseca Sánchez

**Affiliations:** 1Department of Medical Oncology, University Hospital of Salamanca, 37007 Salamanca, Spain; 2Institute for Biomedical Research of Salamanca, 37007 Salamanca, Spain; 3Department of Pathology, University Hospital of Salamanca, 37007 Salamanca, Spain; 4Department of Radiology, University Hospital of Salamanca, 37007 Salamanca, Spain; 5Department of Gastroenterology, University Hospital of Salamanca, 37007 Salamanca, Spain; 6Faculty of Medicine, University of Salamanca, 37007 Salamanca, Spain

**Keywords:** nivolumab, immune checkpoint inhibitors, hepatitis, cholestasis, elderly patient

## Abstract

Immunotherapy has transformed the treatment of many cancers, yet its safety and effectiveness in elderly or frail patients remain uncertain. These individuals may be more susceptible to immune-related toxicities which can be difficult to recognize and manage. In this case report, we describe an older patient treated with nivolumab who developed a severe cholestatic form of immune-mediated hepatitis, a presentation that often responds poorly to standard treatments such as corticosteroids. Despite early recognition and timely immunosuppression, the patient showed only partial improvement and later developed serious complications related to the intensity of therapy. This case raises important questions about how to balance the potential benefits and risks of immunotherapy in vulnerable patients and highlights the need for more research to guide safer and more personalized treatment decisions. As more patients receive these therapies, additional immune-mediated effects will likely emerge, and managing hepatic toxicity may increasingly require rapid, coordinated, multidisciplinary care.

## 1. Introduction

Since its approval in 2014, nivolumab has become one of the most widely used immunotherapy agents in medical oncology. Its mechanism of action is based on the inhibition of the immune checkpoint receptor programmed death-1 (PD-1), expressed on activated T lymphocytes. The binding of PD-1 to its ligands, PD-L1 or PD-L2, delivers inhibitory signals that suppress T-cell proliferation, cytokine production, and cytotoxic activity. This physiological pathway plays a crucial role in maintaining immune tolerance and preventing autoimmune phenomena. However, many tumors exploit this mechanism through PD-L1 overexpression, allowing them to evade immune surveillance.

By blocking the PD-1/PD-L1 interaction, nivolumab restores T-cell activation, enhances antitumor immunity, and promotes the destruction of neoplastic cells [1]. The introduction of immune checkpoint inhibitors (ICIs) like nivolumab has transformed the management of several malignancies—such as metastatic melanoma and non-small cell lung cancer—by significantly improving overall survival. Nevertheless, their use is associated with a distinct spectrum of immune-related adverse events (irAEs), which differ from those typically seen with conventional chemotherapy [2].

Most irAEs are mild and self-limited, with hypothyroidism being the most common. In contrast, hepatic toxicity occurs in approximately 1–6% of patients treated with nivolumab, according to recent series and systematic reviews [3,4]. Hepatic irAEs can manifest with a broader range of phenotypes. Although initially described mainly as hepatocellular injury, more recent cohort studies have shown that cholestatic and mixed patterns—typically marked by disproportionate elevations in ALP and GGT—are increasingly recognized, particularly in patients receiving anti-PD-1 or anti-PD-L1 agents. The hepatocellular injury usually is characterized by asymptomatic elevations in transaminases that resolve after corticosteroid therapy and temporary or permanent discontinuation of the drug [2]. However, the cholestatic and mixed presentations, often with associated biliary involvement are more likely to show a prolonged or even potentially fatal course [5,6].

Here, we report the case of a patient who developed a cholestatic-predominant immune-mediated hepatitis secondary to nivolumab, clinically remarkable for its refractoriness to corticosteroids and only minimal biochemical improvement under mycophenolate mofetil. The patient subsequently died one month later from influenza A infection, a complication that may have been favored by the combined immunosuppressive therapy.

## 2. Case Presentation

An 87-year-old man with a history of scalp melanoma (pT4bN1aM0, diagnosed in July 2018) underwent wide local excision with sentinel lymph node biopsy, followed by clinical surveillance. He remained disease-free until March 2022, when a solitary right frontal lobe brain metastasis was identified and treated with fractionated stereotactic radiotherapy.

In January 2024, the same lesion showed radiologic progression, and the patient was considered unsuitable for additional local therapy. Systemic treatment with dabrafenib plus trametinib was initiated, resulting in temporary disease stabilization. However, in November 2024, further progression of the frontal lesion was documented, and nivolumab monotherapy was commenced as a subsequent-line treatment at a dose of 240 mg intravenously every 2 weeks.

The patient completed 17 cycles of nivolumab with stable disease and no prior immune-related adverse events. On 3 July 2025, prior to the 18th nivolumab infusion, routine blood tests revealed normal liver function (ALT 36 U/L, ALP 68 U/L, GGT 25 U/L, total bilirubin 0.7 mg/dL). The 18th dose was administered on 7 July 2025. The subsequent evolution of liver function parameters is detailed in Table 1 and illustrated in Figure 1.

Three weeks later (28 July 2025), the patient presented with fatigue, anorexia, and mild jaundice. He also reported intense generalized pruritus. Laboratory tests showed a mixed but predominantly cholestatic pattern of liver injury (ALT 269 U/L, AST 191 U/L, ALP 394 U/L, GGT 927 U/L, total bilirubin 2.2 mg/dL, direct bilirubin 1.3 mg/dL). We systematically ruled out the main causes of liver enzyme abnormalities, including viral hepatitis (hepatitis A, B, C and E), autoimmune liver disease (with negative autoantibody screening—ANA, SMA, LKM-1, AMA and SLA—and normal serum IgG levels), cancer progression, vascular complications and other potential drug-related hepatotoxicities. An abdominopelvic ultrasound demonstrated microlithiasis and biliary sludge without bile duct dilatation or other acute findings. Nivolumab was discontinued, and prednisone 1 mg/kg/day was initiated for suspected immune-mediated hepatitis.

Despite corticosteroid therapy, liver enzymes worsened (6 August 2025): ALT 585 U/L, AST 302 U/L, ALP 526 U/L, GGT 2261 U/L, total bilirubin 2.0 mg/dL. A CT thorax–abdomen–pelvis excluded hepatic metastases or biliary obstruction. Given the lack of response, mycophenolate mofetil (1 g twice daily) was added to corticosteroids, and ursodeoxycholic acid was initiated.

A liver biopsy performed on 13 August 2025 revealed preserved hepatic architecture with mild portal inflammation, focal lobular necrosis, and intracanalicular cholestasis, without fibrosis or ductular proliferation—findings consistent with lobular-predominant hepatitis compatible with immune-mediated, drug-induced injury (Ishak score 5) (Figure 2).

During the first days of mycophenolate therapy, a transient biochemical flare was observed (20 August 2025), with total bilirubin peaking at 4.7 mg/dL (direct 3.7 mg/dL), ALP 214 U/L, and GGT 1195 U/L. A magnetic resonance cholangiopancreatography performed the same day showed normal intrahepatic and extrahepatic bile ducts with no obstruction (Figure 3 and Figure 4). Over the following week, a partial biochemical improvement occurred (27 August 2025): ALT 222 U/L, AST 96 U/L, ALP 481 U/L, GGT 1758 U/L, total bilirubin 1.9 mg/dL.

Despite partial recovery, the patient experienced progressive frailty and was referred to the Palliative Care Unit. He died on 24 September 2025, from influenza A pneumonia, considered unrelated to the hepatic event.

A summary of liver function trends is presented in Table 1, and imaging and histopathologic findings in Table 2.

## 3. Discussion

Immune checkpoint inhibitor (ICI)-induced liver toxicity is characterized by a broad and heterogeneous spectrum of clinical and pathological presentations. Using the R ratio, calculated as (ALT/ULN)/(ALP/ULN), ICI-related hepatotoxicity can be classified into three major patterns: hepatocellular, cholestatic and mixed hepatitis. Hepatocellular liver function test abnormalities tend to occur earlier and are more frequently associated with anti-CTLA-4 therapy, whereas cholestatic and cholangitic patterns—characterized by disproportionate elevations of ALP and GGT—are more commonly observed in patients treated with anti-PD-1 or anti-PD-L1 agents and usually exhibit a longer latency. Early descriptions focused mainly on hepatocellular patterns, but accumulating evidence from larger cohorts demonstrates that cholestatic and mixed phenotypes are clinically frequent and biologically relevant. In a multicenter series of 117 patients, cholestatic and mixed presentations accounted for more than half of ICI-induced liver injuries and were often accompanied by biliary involvement [6].

Similar findings were reported in a 213-patient real-world cohort from Korea, where 35.4% of cases exhibited a cholestatic profile. Collectively, these data confirm that cholestatic-predominant liver injury is a well-recognized and clinically meaningful phenotype within ICI-related hepatotoxicity [7].

By presenting this case, we aim to increase awareness among clinicians, as cholestatic or cholangitic patterns of immune-mediated liver injury may be underappreciated in routine practice and are frequently associated with poorer response to corticosteroids and greater therapeutic refractoriness. This has been consistently observed across observational cohorts, in which cholestatic-dominant injury shows slower biochemical improvement, prolonged immunosuppression requirements and a higher need for second-line therapies compared with hepatocellular patterns. In this context, our case illustrates this more severe end of the spectrum: despite early initiation of corticosteroids, the patient showed limited biochemical response and required prolonged immunosuppression.

Cholestasis refers to a partial or complete impairment of bile formation or flow, resulting in the reduction or absence of bile reaching the duodenum. This alteration may arise from a broad spectrum of diseases that share overlapping clinical, analytical, and histopathological manifestations. In the diagnostic evaluation of immune-related hepatitis with cholestatic features, it is essential to exclude alternative causes such as congestive heart failure, drug-induced liver injury, or mechanical obstruction of the biliary tract [8].

Differentiating intrahepatic from extrahepatic cholestasis is crucial. In cases of intrahepatic cholestasis without ductal obstruction, as in the present case, hepatic metastases must also be ruled out before attributing the injury to nivolumab. Although diffuse hepatic infiltration can produce similar biochemical alterations, frank jaundice is uncommon unless massive parenchymal replacement occurs, supporting the utility of MRCP in the differential diagnosis [9].

Before exploring the underlying pathophysiology, it is important to distinguish immune checkpoint inhibitor-induced hepatitis from classic autoimmune hepatitis (AIH), as both may present with similar biochemical profiles but differ substantially in their immunologic, histologic, and clinical characteristics. AIH typically presents with high-titer autoantibodies (ANA, SMA, LKM-1), elevated IgG levels, and interface hepatitis with plasma-cell-rich infiltrates. In contrast, ICI hepatitis usually lacks disease-specific autoantibodies, displays normal IgG levels, and shows a predominantly lobular pattern with CD8^+^ T-cell-mediated injury and minimal plasma-cell infiltrates. Clinically, AIH follows a chronic, relapsing course requiring long-term immunosuppression, while ICI-induced hepatitis is usually monophasic and rarely relapses after discontinuation of the checkpoint inhibitor [10].

To help distinguish AIH from other liver diseases, the International Autoimmune Hepatitis Group (IAIHG) proposed specific diagnostic criteria and scoring tools. The first scoring system was introduced in 1993, revised in 1999, and later complemented by simplified criteria in 2008. Although the simplified score performs well in classical AIH, its diagnostic accuracy decreases in acute hepatitis, drug-induced autoimmune-like hepatitis (DI-ALH), and in the presence of concomitant liver diseases—contexts in which relying solely on the simplified system may be misleading. Our case lies precisely within these limitations. Therefore, we applied the Revised Original IAIHG Score (1999), which incorporates drug exposure (−4 points for hepatotoxic agents). The resulting score was 4 points, classifying AIH as “improbable,” reflecting negative autoimmune serology, normal IgG levels, the absence of typical AIH histology, and the recent use of nivolumab, a recognized trigger of immune-mediated but not autoimmune hepatitis [11].

Although cholestatic patterns of AIH have been described historically [12], cholestasis is no longer considered a phenotypic variant of AIH according to the EASL Clinical Practice Guidelines 2025, where its presence generally suggests an overlap syndrome (AIH–PBC or AIH–PSC) or another concomitant liver disease rather than a distinct AIH subtype [13]. In contrast, cholestatic or mixed hepatitic–cholestatic patterns are well-recognized within the spectrum of ICI-mediated liver injury and are frequently associated with poorer response to corticosteroids, slower biochemical improvement, and a greater need for second-line immunosuppression—features consistent with the clinical behavior observed in our patient.

These differences support the concept that immune checkpoint inhibitor-induced hepatotoxicity represents a distinct, transient immune-mediated process rather than a classic autoimmune disease, despite partial biochemical and histologic overlap.

The pathophysiology of ICI-induced liver injury remains incompletely understood, largely due to the limited number of biopsies performed, given the rapid resolution of most cases and the invasive nature of the procedure. The most accepted mechanism involves T-cell-mediated autoimmunity against hepatocytes triggered by the loss of immune tolerance secondary to PD-1/PD-L1 pathway blockade [3,10]. Experimental studies and small human series have demonstrated upregulation of PD-L1 in hepatocytes, Kupffer cells, and biliary epithelial cells during inflammatory states, suggesting that the PD-1/PD-L1 axis plays a central role in maintaining hepatic immune homeostasis. Its pharmacologic inhibition may thus unmask cytotoxic T-cell activity against hepatocytes and canalicular structures, explaining the mixed hepatocellular–cholestatic damage observed in some cases [14].

Although immune-mediated injury against bile acid transporters (such as BSEP or MDR3) has been postulated in other cholestatic drug reactions, there is currently no direct evidence of such autoimmunity in the context of ICI-related liver injury. The available data support a primarily hepatocellular or canalicular mechanism rather than true bile duct pathology [15].

Liver biopsy is recommended in the most severe cases (grade 3 and 4 according to the Common Terminology Criteria for Adverse Events, CTCAE v5.0), to confirm the diagnosis and to rule out an unrecognized chronic liver disease. Histologic findings in immune-mediated hepatitis (IMH) are heterogeneous, and no single pathognomonic feature has been identified. Peeraphatdit et al. described two predominant histologic patterns: (1) acute hepatitis with lobular inflammation and acidophil bodies, and (2) centrilobular necrosis [3]. Similarly, Zhang et al., in what remains the most comprehensive morphologic study to date, observed acute lobular hepatitis in most patients, with spotty necrosis and acidophil bodies in the majority and centrilobular necrosis in only one case [16].

In our patient, the liver biopsy revealed acute lobular hepatitis with multiple foci of lytic (focal) necrosis and marked intracanalicular cholestasis, but no bile duct injury or ductular proliferation. This combination defines an intermediate mixed pattern with cholestatic predominance, positioned within the hepatocellular spectrum of ICI-induced injury rather than a cholangitic phenotype. The presence of focal lobular necrosis without confluent collapse supports an active immune-mediated process with partial hepatocellular injury and canalicular involvement.

At the onset of hepatitis, the calculated R-ratio was approximately 2.0, indicating a mixed pattern with clear cholestatic predominance and lying at the threshold of a purely cholestatic injury. This biochemical profile, together with the histological finding of intracanalicular cholestasis and mild bile duct injury, supports the diagnosis of cholestatic-predominant immune-mediated hepatitis.

Taken together, these findings align with the lobular patterns of IMH described in the literature but highlight an uncommon cholestatic-predominant variant, seldom illustrated and often associated with a slower and corticosteroid-refractory course, as seen in our patient. This emphasizes the clinical importance of recognizing such phenotypes, where early introduction of second-line immunosuppressants such as mycophenolate may facilitate biochemical recovery.

In our case, however, only a minimal biochemical response was observed after the initiation of mycophenolate mofetil, reflecting the limited reversibility that sometimes characterizes severe cholestatic forms of immune-mediated hepatitis. Unfortunately, the patient died one month later from influenza A infection, a complication that may have been favored by the combined immunosuppressive therapy. This adverse outcome highlights the challenges of managing immune-related hepatotoxicity in elderly and frail patients, who often have reduced physiological reserves, multiple comorbidities, and increased susceptibility to infectious complications. In this population, the therapeutic balance between suppressing immune-mediated injury and preserving immune competence becomes particularly delicate. Careful risk–benefit assessment, close monitoring for infections, and preventive measures such as seasonal vaccination are essential to minimize treatment-related morbidity and mortality.

## 4. Conclusions

This case illustrates a presentation of cholestatic-predominant immune-mediated hepatitis induced by nivolumab, confirmed histologically as lobular hepatitis with intracanalicular cholestasis. Recognition of this phenotype is crucial, as it may follow a slow, corticosteroid-refractory course and require early introduction of second-line immunosuppressants such as mycophenolate mofetil. However, in elderly or frail patients, the use of intensive immunosuppression must be weighed against the heightened risk of severe infectious complications, underscoring the need for individualized management, vigilant monitoring, and preventive strategies to minimize morbidity and mortality.

## Figures and Tables

**Figure 1 curroncol-32-00663-f001:**
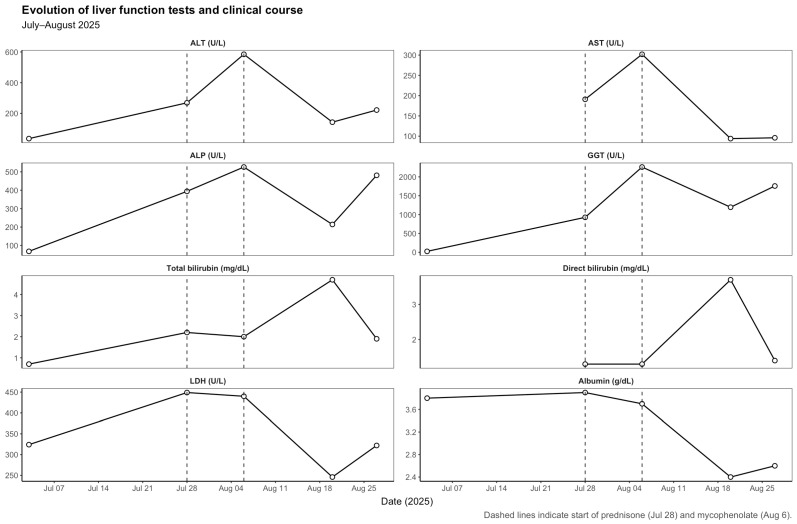
Evolution of liver enzymes and biochemical parameters during the course of steroid-refractory immune-mediated hepatitis (July–August 2025).

**Figure 2 curroncol-32-00663-f002:**
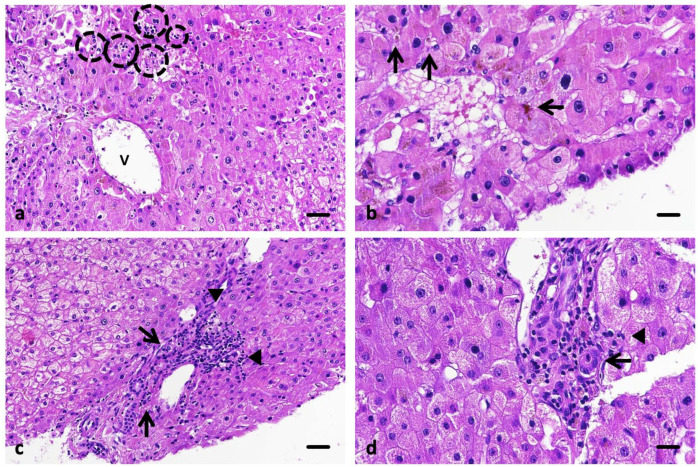
Histological findings in the biopsy. In image (**a**) are depicted several foci of spotty necrosis (discontinuous circles) surrounding a central vein (v). Image (**b**) demonstrates foci of biliary cholestasis (arrow). Images (**c**,**d**) illustrate ductular damage (arrow) and light portal inflammation with scarce interface hepatitis (arrowheads). Scale bar 50 µm (**a**,**c**), 25 µm (**b**,**d**).

**Figure 3 curroncol-32-00663-f003:**
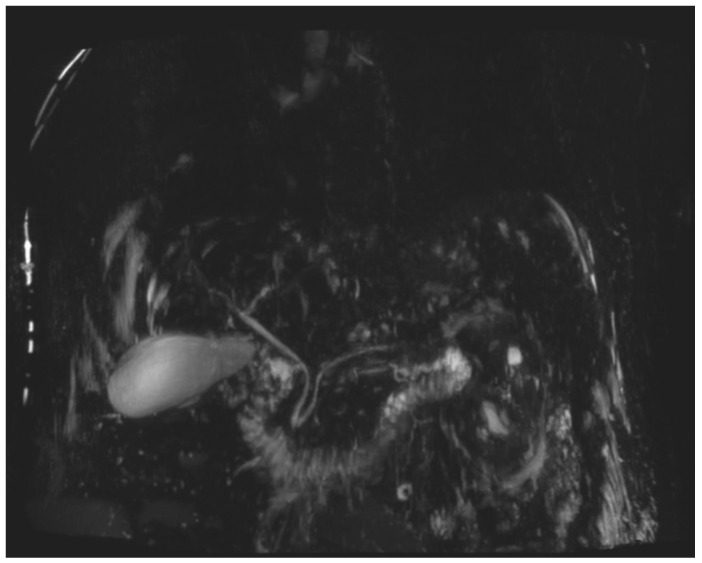
Three-dimensional MR cholangiopancreatography (3D-MRCP). The intrahepatic and extrahepatic bile ducts appear normal in caliber and course, with no evidence of stenosis or dilatation.

**Figure 4 curroncol-32-00663-f004:**
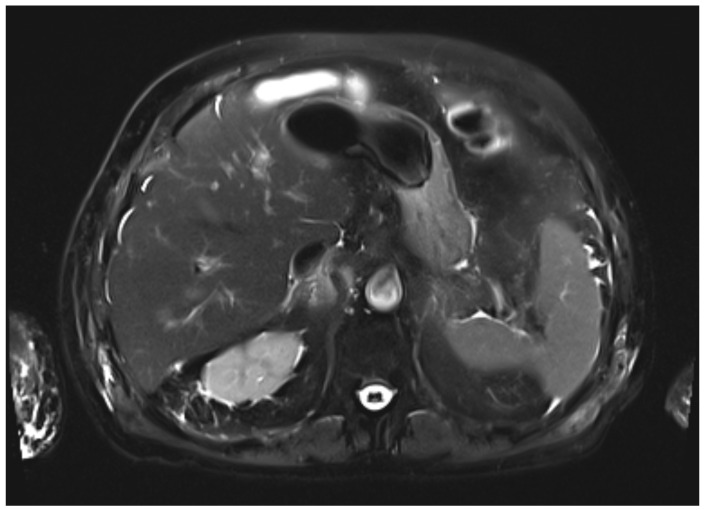
Axial T2-weighted HASTE MR image. The liver parenchyma shows homogeneous signal intensity, without focal lesions or signs of cholestasis.

**Table 1 curroncol-32-00663-t001:** Evolution of liver function tests and clinical course (July–August 2025).

Date (2025)	Clinical Context	ALT (U/L)	AST (U/L)	ALP (U/L)	GGT (U/L)	Total Bilirubin (mg/dL)	Direct Bilirubin (mg/dL)	LDH (U/L)	Albumin (g/dL)	Comments
July 3	On nivolumab (cycle 17), normal liver tests	36	–	68	25	0.7	–	324	3.8	Normal profile before last nivolumab cycle
July 28	Onset of immune-mediated hepatitis, prednisone started	269	191	394	927	2.2	1.3	449	3.9	Mixed but predominantly cholestatic hepatitis (grade 3–4)
August 6	Worsening under corticosteroids, mycophenolate initiated	585	302	526	2261	2.0	1.3	440	3.7	Severe steroid-refractory cholestatic hepatitis
August 20	Peak under mycophenolate (biochemical flare)	143	94	214	1195	4.7	3.7	246	2.4	Marked cholestasis and hyperbilirubinemia, transient worsening
August 27	Partial biochemical improvement	222	96	481	1758	1.9	1.4	322	2.6	Partial recovery under continued mycophenolate; persistent cholestasis
September 24	Palliative care; death from influenza A pneumonia	–	–	–	–	–	–	–	–	Death unrelated to immune hepatitis

**Table 2 curroncol-32-00663-t002:** Chronological summary of diagnostic studies and key findings during the evaluation of immune-mediated hepatitis.

Date (2025)	Diagnostic Test	Main Findings	Interpretation
July 28	Abdominopelvic ultrasound	Mildly heterogeneous hepatic echotexture. Gallbladder with microlithiasis and sludge, no wall thickening, no intra- or extrahepatic bile duct dilatation.	No biliary obstruction or urgent findings.
August 7	CT Thorax–Abdomen–Pelvis	Stable right frontal brain metastasis. No hepatic lesions or biliary dilatation. No evidence of disease progression.	Excludes hepatic metastases or obstructive pathology.
August 13	Liver biopsy	Preserved hepatic architecture; mild portal inflammation without interface activity or fibrosis; lobular necrosis (Ishak score 5) with intracanalicular cholestasis; no ductular proliferation; <2% steatosis.	Lobular-predominant hepatitis with focal cholestasis, compatible with drug-induced (immune-mediated) hepatitis.
August 20	MR cholangiopancreatography	Normal intra- and extrahepatic biliary ducts, no strictures or dilatation; liver and gallbladder unremarkable; only small renal cysts.	Normal biliary anatomy, no obstruction.

## Data Availability

Data are available on request from the corresponding author. The data are not publicly available due to privacy or ethical restrictions. No data that could compromise patient privacy will be shared.

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
