# Peer review of "Steroid-Refractory Cholestatic Immune-Mediated Hepatitis Following Nivolumab Therapy in an Elderly Patient with Metastatic Melanoma: A Rare and Challenging Presentation"

_curroncol, 2025, doi:10.3390/curroncol32120663_

Round 1

Reviewer 1 Report

Comments and Suggestions for Authors

The manuscript presents an interesting case of steroid-refractory, cholestatic-predominant immune-related hepatitis, which is clinically relevant and contributes to the understanding of uncommon toxicities associated with PD-1 blockade. However, I would like to raise two points that may require clarification or additional discussion:

1) Frequency of cholestatic-predominant liver injury

The authors describe cholestatic-predominant immune-related hepatitis as “very rare,” based mainly on the limited number of published case reports identified through a PubMed search. In clinical practice, however, mixed or cholestatic-predominant liver injury patterns are observed in a non-negligible proportion of patients receiving immune checkpoint inhibitors. Relying solely on case reports may underestimate the actual frequency, and incorporating data from cohort studies or larger series would provide a more balanced perspective. Clarifying this point could strengthen the manuscript’s discussion of the case’s uniqueness.

2) Interpretation of the MRCP

The manuscript states that the MRCP was normal and that biliary involvement was excluded. However, the provided images appear to have limited resolution, making it difficult to confidently rule out immune-related cholangitis. In ICI-associated cholangitis, subtle bile duct wall thickening or irregularity can be more prominent than strict stenosis or ductal dilatation, and these findings may be missed on lower-resolution sequences. If available, higher-resolution images or a more detailed radiologic interpretation would support the conclusion that immune-related cholangitis was not present.

Addressing these points would further clarify the diagnostic reasoning and enhance the manuscript’s overall robustness.

Author Response

REVIEWER 1

The manuscript presents an interesting case of steroid-refractory, cholestatic-predominant immune-related hepatitis, which is clinically relevant and contributes to the understanding of uncommon toxicities associated with PD-1 blockade. However, I would like to raise two points that may require clarification or additional discussion:

1) Frequency of cholestatic-predominant liver injury

The authors describe cholestatic-predominant immune-related hepatitis as “very rare,” based mainly on the limited number of published case reports identified through a PubMed search. In clinical practice, however, mixed or cholestatic-predominant liver injury patterns are observed in a non-negligible proportion of patients receiving immune checkpoint inhibitors. Relying solely on case reports may underestimate the actual frequency, and incorporating data from cohort studies or larger series would provide a more balanced perspective. Clarifying this point could strengthen the manuscript’s discussion of the case’s uniqueness.

We thank the reviewer for this important observation. We agree that describing cholestatic-predominant ICI-related liver injury as “very rare” based solely on case reports underestimates its true frequency. As suggested, we have removed the term rare from lines 38 and 51.

Furthermore, we substantially revised the Discussion to incorporate evidence from cohort studies demonstrating that cholestatic and mixed patterns represent a meaningful proportion of ICI-induced liver injury. Specifically, in a multicentre cohort of 117 patients, cholestatic or mixed presentations accounted for over half of cases, often with associated biliary involvement (Ref. 1). Similar findings were observed in a 213-patient real-world Korean cohort, where 35.4% of cases showed a cholestatic profile (Ref. 2).

To address this point thoroughly, we deleted the paragraph originally spanning lines 84–86 and replaced it with the following new text:

“However, immune checkpoint inhibitor–induced liver injury can manifest with a broader range of phenotypes. Although initially described mainly as hepatocellular injury, more recent cohort studies have shown that cholestatic and mixed patterns—typically marked by disproportionate elevations in ALP and GGT—are increasingly recognized, particularly in patients receiving anti-PD-1 or anti-PD-L1 agents. Notably, these cholestatic and mixed presentations may account for more than half of ICI-related liver injuries, often with associated biliary involvement and showing a prolonged or even potentially fatal course.”

In the Discussion section (lines 168–183), we also replaced the introductory paragraph with a new, more comprehensive summary reflecting the updated evidence. This revised introduction integrates the reviewer’s observation and highlights that cholestatic presentations are clinically frequent and biologically relevant, rather than rare or exceptional.

These changes provide a more accurate and balanced representation of the frequency of cholestatic-predominant injury in clinical practice.

2) Interpretation of the MRCP

The manuscript states that the MRCP was normal and that biliary involvement was excluded. However, the provided images appear to have limited resolution, making it difficult to confidently rule out immune-related cholangitis. In ICI-associated cholangitis, subtle bile duct wall thickening or irregularity can be more prominent than strict stenosis or ductal dilatation, and these findings may be missed on lower-resolution sequences. If available, higher-resolution images or a more detailed radiologic interpretation would support the conclusion that immune-related cholangitis was not present.

Addressing these points would further clarify the diagnostic reasoning and enhance the manuscript’s overall robustness.

We thank the reviewer for this insightful and clinically relevant comment. Following the reviewer’s suggestion, the MRCP images were re-evaluated by an experienced abdominal radiologist. Particular attention was given to subtle findings associated with ICI-related cholangitis, including bile duct wall thickening, irregularity, and focal caliber changes—features that may precede ductal dilatation or strict stenosis.

Upon detailed re-review, no abnormalities suggestive of immune-related cholangitis were identified. The intra- and extrahepatic bile ducts were normal in contour, caliber, and wall thickness. Additionally, the liver biopsy showed no ductular reaction, ductopenia, or destructive cholangitis. Together, these radiological and histological findings support the conclusion that immune-related cholangitis was not present in this case.

Importantly, immune-related cholangitis was one of our initial diagnostic suspicions, particularly because the patient presented with a clear cholestatic biochemical profile and experienced minimal response to corticosteroids—features commonly associated with the cholangitic phenotype of ICI hepatotoxicity. For this reason, special emphasis was placed on carefully reviewing the MRCP and correlating the imaging with the histological findings.

To reinforce this diagnostic reasoning, we have expanded both the Case Presentation and the Discussion. The revised text now explains in more detail why cholangitis was initially considered, how it was systematically evaluated radiologically and histologically, and why it was ultimately excluded.

We also improved the radiologic description in the manuscript and enhanced image quality where technically feasible.

Reviewer 2 Report

Comments and Suggestions for Authors

In this manuscript, Dr. Luis Posado Domínguez and Colleagues report the case of an 87-year-old man with a history of scalp metastatic melanoma who developed a cholestatic-predominant immune-mediated hepatitis in the course of treatment with nivolumab, characterized by refractoriness to corticosteroids and only minimal biochemical improvement under mycophenolate mofetil. Liver biopsy showed acute lobular hepatitis with intracanalicular cholestasis and mild bile duct injury, consistent with immune-mediated, drug-induced injury (Ishak score 5). Mycophenolate mofetil produced only partial biochemical improvement. The authors discussed this rare cholestatic-predominant phenotype of nivolumab-induced hepatitis, characterized by poor treatment response and incomplete recovery despite second-line immunosuppression. The manuscript has clinical interest. However, in my opinion, the clinical impact could be improved by addressing some important issues.

-diagnostic workup: in July 2025, after persistent liver enzyme elevation, the authors report only viral markers and abdominal ultrasound as diagnostic procedures, and then corticosteroid and UDCA treatments were introduced. In August, liver biopsy was performed under treatment. Could the ongoing treatment affect the histological assessment? In my opinion, the authors should also describe whether autoantibody screening was also performed, since any newly introduced drug, including immune checkpoint inhibitors, might act as a trigger of autoimmune liver diseases. Although less frequent, a cholestatic form of autoimmune hepatitis has been reported in the literature (doi: 10.1016/j.cgh.2013.08.039.) and, to exclude this diagnosis, it would be useful to report in the manuscript the diagnostic scores of autoimmune hepatitis, particularly the roginal revised score (1999) that, differently form the somplified one (2008), consider "drug history" a parameter giving a negative score (-4), thus providing a more useful diagnostic significance in excluding AIH in such a clinical setting, as well described in a comprehensive review addressing the main differences between the original-revised and the simplified scores (DOI: 10.2174/138955709788452676), and discussing both conventional and supplemental autoantibody repertoire that should be performed to accurately exclude AIH diagnosis. The relevance of such a careful diagnostic work-up relies in the duration of immunosuppressive treatment that is long-life in AIH while is temporary in a drug-induced liver injury.

Author Response

In this manuscript, Dr. Luis Posado Domínguez and Colleagues report the case of an 87-year-old man with a history of scalp metastatic melanoma who developed a cholestatic-predominant immune-mediated hepatitis in the course of treatment with nivolumab, characterized by refractoriness to corticosteroids and only minimal biochemical improvement under mycophenolate mofetil. Liver biopsy showed acute lobular hepatitis with intracanalicular cholestasis and mild bile duct injury, consistent with immune-mediated, drug-induced injury (Ishak score 5). Mycophenolate mofetil produced only partial biochemical improvement. The authors discussed this rare cholestatic-predominant phenotype of nivolumab-induced hepatitis, characterized by poor treatment response and incomplete recovery despite second-line immunosuppression. The manuscript has clinical interest. However, in my opinion, the clinical impact could be improved by addressing some important issues.

-diagnostic workup: in July 2025, after persistent liver enzyme elevation, the authors report only viral markers and abdominal ultrasound as diagnostic procedures, and then corticosteroid and UDCA treatments were introduced. In August, liver biopsy was performed under treatment. Could the ongoing treatment affect the histological assessment? In my opinion, the authors should also describe whether autoantibody screening was also performed, since any newly introduced drug, including immune checkpoint inhibitors, might act as a trigger of autoimmune liver diseases. Although less frequent, a cholestatic form of autoimmune hepatitis has been reported in the literature (doi: 10.1016/j.cgh.2013.08.039.) and, to exclude this diagnosis, it would be useful to report in the manuscript the diagnostic scores of autoimmune hepatitis, particularly the roginal revised score (1999) that, differently form the somplified one (2008), consider "drug history" a parameter giving a negative score (-4), thus providing a more useful diagnostic significance in excluding AIH in such a clinical setting, as well described in a comprehensive review addressing the main differences between the original-revised and the simplified scores (DOI: 10.2174/138955709788452676), and discussing both conventional and supplemental autoantibody repertoire that should be performed to accurately exclude AIH diagnosis. The relevance of such a careful diagnostic work-up relies in the duration of immunosuppressive treatment that is long-life in AIH while is temporary in a drug-induced liver injury.

We thank the reviewer for this insightful and clinically relevant comment. As the reviewer correctly notes, cholestatic presentations of autoimmune hepatitis (AIH) have been described in the classical literature, including the review by Czaja (Clin Gastroenterol Hepatol 2014), which proposed several “cholestatic phenotypes” of AIH. We acknowledge this earlier conceptual framework and have now incorporated this perspective into the revised manuscript.

However, according to the most recent EASL Clinical Practice Guidelines (2024–2025), such cholestatic presentations are no longer considered phenotypic variants within AIH. Instead, patients who present with cholestatic biochemical or histological abnormalities together with autoimmune features are now classified under distinct diagnostic entities, including AIH–PBC overlap, AIH–PSC overlap, AMA-negative PBC, or small-duct PSC. Therefore, cholestasis alone is not regarded as a presentation of AIH in contemporary diagnostic criteria.

Round 2

Reviewer 1 Report

Comments and Suggestions for Authors

The authors have adequately addressed all previous comments, and the revised manuscript is now clear, accurate, and well-supported by the literature. The discussion has been strengthened, and the diagnostic reasoning is coherent. I have no further concerns. The manuscript is suitable for publication.